# Proteomic Analysis of the *Fusarium graminearum* Secretory Proteins in Wheat Apoplast Reveals a Cell-Death-Inducing M43 Peptidase

**DOI:** 10.3390/jof11040240

**Published:** 2025-03-21

**Authors:** Pengfeng Li, Ruihua Zhao, Ying Fang, Yujin Fan, Qianyong Hu, Wei Huang, Wujun Ma, Cuijun Zhang

**Affiliations:** 1Shenzhen Branch, Guangdong Laboratory of Lingnan Modern Agriculture, Key Laboratory of Synthetic Biology, Ministry of Agriculture and Rural Affairs, Agricultural Genomics Institute at Shenzhen, Chinese Academy of Agricultural Sciences, Shenzhen 518120, China; pengfengli17@126.com (P.L.); ruihuazhao1212@163.com (R.Z.); 17319781516@163.com (Y.F.); 82101215165@caas.cn (Q.H.); 2College of Life Sciences, South China Agricultural University, Guangzhou 510642, China; weihuang@scau.edu.cn; 3Centre for Crop and Food Innovation, Food Futures Institute, School of Agriculture Science, Murdoch University, Perth 6150, Australia; 4College of Life Science and Technology, Huazhong Agricultural University, Wuhan 430070, China; fy18907941825@126.com; 5College of Agronomy, Qingdao Agricultural University, Qingdao 266109, China

**Keywords:** *Fusarium graminearum*, wheat, secretory proteins, M43 peptidase

## Abstract

*Fusarium graminearum*, a highly destructive fungal pathogen, poses a major threat to wheat production. The apoplast is an important space for plant–pathogen interactions. However, no studies have been reported on the secretory proteins of *F. graminearum* in the wheat apoplast. In this study, we performed mass spectrometry analysis of *F. graminearum* secretory proteins in wheat apoplast and identified 79 potential secretory proteins. We identified a metalloprotease (referred to as Fg28) and demonstrated its capacity to induce cell death and reactive oxygen species (ROS) accumulation in *Nicotiana benthamiana*. Fg28 is strongly up-regulated in the early stages of infection and is secreted into the intercellular space of wheat cells. Full-length Fg28 is required to induce cell death in *N. benthamiana*. In addition, Fg28 induces an immune response that is independent of BAK1/SOBIR1 and EDS1/PAD4. Furthermore, knocking out *Fg28* had no effect on morphology or pathogenicity. In conclusion, we have identified a set of *F. graminearum* secreted proteins in the wheat apoplast and a metalloproteinase that triggers immune response, providing new insights into understanding the interaction between *F. graminearum* and wheat.

## 1. Introduction

Wheat is a major food crop worldwide, but it is frequently threatened by a variety of diseases, including Fusarium crown rot, Fusarium head blight (FHB), rust and powdery mildew. *Fusarium graminearum*, a hemi-biotrophic pathogen, is the major cause of FHB and Fusarium crown rot, and the deoxynivalenol toxin (DON) it secretes is a very serious threat to wheat food safety [1,2,3,4]. Therefore, understanding the pathogenic mechanism of *F. graminearum* will provide insights for breeding and improving FHB resistance in wheat.

*F. graminearum* interferes with wheat immunity by secreting proteins and toxins (DON) during infection [5,6]. A number of secreted proteins have been identified in *F. graminearum*, including pathogen-associated molecular patterns (PAMPs), effectors and toxic proteins. OSP24, as a cytoplasmic effector, inhibits BAX- and INF1-induced cell death and competes with TaFROG for interaction with TaSnRK1α, thereby accelerating the degradation of TaSnRK1α and inhibiting immune responses in wheat [7]. Toxic proteins also play an important role in the pathogenicity of *F. graminearum*. Previously, proteomic analyses of CMC medium identified five subtilisin-like proteases (FgSLPs) in *F. graminearum* that induce cell death in *Nicotiana benthamiana (N. benthamiana)* leaves. Knockout mutants of two of these proteases, FgSLP1 and FgSLP2, reduce *F. graminearum* virulence, indicating the importance of secreted toxic proteins in *F. graminearum* pathogenicity [8]. In addition, several secreted proteins have also been identified in *F. graminearum* that affect pathogenicity, including FgEC1, FGL1 and FGL15 (two lipases), the salicylate hydroxylase gene FgNahG, the secreted ribonuclease Fg12 and the extracellular effector Fg62 [6,9,10,11,12,13,14,15].

The plant apoplast is the primary space of plant–pathogen interactions [16,17]. The apoplast contains pathogen- and plant-secreted signaling molecules (e.g., Flg22 and chitin) and proteins, with the capacity to influence the immune response. The fungal pathogen *Fusarium oxysporum* secretes rapid alkalinization factor (RALF)-like effectors, which alter apoplast pH and thereby enhance pathogen infection [18]. PsXEG1 is a major glycoside hydrolase of *Phytophthora sojae* and functions as a PAMP in the apoplast. PsXEG1 contributes to virulence depends on its xyloglucanase activity that damages soybean cell wall; however, its function is inhibited by GmGIP1. At the same time, XEG1 is recognized by RXEG1, resulting in the inhibition of its glycoside hydrolase activity [19]. Two other immune-triggering apoplast proteins, AEP1 (aldose 1-epimerase1) and PsCAP1 (cysteine-rich secretory protein, antigen 5 and pathogenesis-related 1 protein), have also been identified in the apoplast fluid [20,21]. This suggests that secretory proteins of the pathogen in the apoplast are important for infection of plants. However, the secretory proteins of *F. graminearum* in the wheat apoplast remain largely unknown.

In this study, we performed proteomic analyses of apoplast fluid from wheat seedling coleoptiles inoculated with *F. graminearum* and found that one of the secretory proteins, Fg28 (encoded by *FGSG_00028*), has the capacity to induce cell death and reactive oxygen species (ROS) accumulation. Fg28 was localized to the plasma membrane and the full-length Fg28 protein is essential for its function. Furthermore, Fg28 induced the up-regulation of immune genes in *N. benthamiana*, and Fg28-induced cell death and immune responses were independent of BRI1-ASSOCIATED KINASE-1 (BAK1), SUPPRESSOR OF BIR1-1 (SOBIR1), ENHANCED DISEASE SUSCEPTIBILITY 1 (EDS1) and PHYTOALEXIN DEFICIENT 4 (PAD4). The knockout of *Fg28* exhibited no impact on the pathogenicity of *F. graminearum*. In conclusion, we provide a set of proteomic data from the apoplast fluid of wheat seedlings and have identified a protein that can induce cell death.

## 2. Materials and Methods

### 2.1. Experimental Materials and Conditions for Growing

The wild-type *F. graminearum* strain PH-1 and *Fg28* mutant strains were grown on potato dextrose agar (PDA) medium at 28 °C in the dark. Conidia was obtained by cultivating the fresh mycelium in a carboxymethylcellulose (CMC) medium at 28 °C for 5 d. Wheat seedlings (Fielder) were grown at 22 °C with a 14/10 h (h) day/night cycle for 2 weeks and used for apoplast fluid isolation. Wheat spikes (Fielder), growing in a greenhouse at 22 °C with a 14/10 h day/night cycle, were used for virulence evaluation. *N. benthamiana* (wild type, *bak1*, *sobir1*, *eds1* and *pad4*) was grown in a greenhouse at 22 °C under 16 h light/8 h dark cycle for 4 weeks. *Agrobacterium tumefaciens* strain GV3101 was used for transient expression of target proteins in *N*. *benthamiana* leaves.

### 2.2. Apoplast Fluid Extraction and Liquid Chromatography–Tandem Mass Spectrometry (LC-MS/MS)

Apoplast fluid extraction was performed using the infiltration centrifugation technique as previously described with slight modifications [21,22]. Briefly, one hundred of wheat (Fielder) coleoptiles were inoculated with fresh *F*. *graminearum* conidial suspensions (OD = 1.5) and incubated for 48 h. The coleoptiles were full filled with phosphate buffer saline (PBS) by vacuum pumping and releasing. The apoplast fluid was collected by centrifuged at 4000 rpm for 30 min at 4 °C and concentrated by lyophilization [23,24]. Then, the concentrated apoplast proteins were running into SDS-PAGE gel for proteomics analysis. Target protein bands were enzymatically digested, and the resulting peptides were desalted and resuspended in Nano-HPLC Buffer A. Peptide separation was performed using an EASY-nLC 1200 system with a trap column and a 75 μm × 150 mm analytical column (RP-C18, Thermo Inc., Waltham, MA, USA) at 300 nL/min. A 30 min gradient wash was used between samples to prevent carryover. Mass spectrometry analysis was performed on a Q-Exactive instrument (Thermo Scientific) in data-dependent acquisition (DDA) mode. The full scan range was 300–1500 *m*/*z*, with the top 20 most intense ions selected for HCD fragmentation (NCE 28). Dynamic exclusion was set to 25 s. MS1 resolution was 70,000 (at *m*/*z* 200), and MS2 resolution was 17,500, with AGC targets of 3e6 and 1e5, respectively [25,26]. All experiments were performed three biological repeats.

### 2.3. Bioinformatics Analysis

Signal peptide (SP) prediction was performed using the online SignalP-5.0 online server (https://services.healthtech.dtu.dk/services/SignalP-5.0/, accessed on 10 November 2021) and InterPro (https://www.ebi.ac.uk/interpro/, accessed on 10 November 2021). Protein domain was annotated by InterPro and Uniport. Protein function annotation was extracted from UniProt (https://www.uniprot.org/, accessed on 10 November 2021). Veen and heatmap analysis were performed by R scripts (https://www.r-project.org/, accessed on 15 November 2021). The expression data of 79 candidate gene were downloaded from previous study [27]. KEGG enrichment was performed based on the KEGG database (https://www.kegg.jp/, accessed on 15 November 2021) [28]. A total of 8 homologous protein sequences of Fg28 from *Fusarium* species and 5 of other fungal species were identified by querying the Fg28 protein sequence against the Ensembl Fungi database. Additionally, four metalloprotease sequences were obtained from a previously published study [29,30,31,32]. Multiple sequence alignment of Fg28 and its homologs was performed using MAFFT. A phylogenetic tree was then constructed using the Neighbor-Joining (NJ) method in MEGA 5 (www.megasoftware.net, accessed on 10 March 2025).

### 2.4. Agrobacterium-Mediated Transient Expression in N. benthamiana

The coding sequence of the Fg28 and its mutants (Fg28^ΔM43^, Fg28^ΔSP^, Fg28^SP+M43^, Fg28^ΔC200^ and Fg28^ΔC400^) were amplified from wild-type *F*. *graminearum* cDNA, and then the purified fragments were cloned into pCambia1300-Flag plasmid using the ClonExpress II One Step Cloning Kit (Vazyme Biotech Co. Ltd., Nanjing, China). The primers used in this study are listed in Appendix A. The plasmids were transformed into *A. tumefaciens* strain GV3101 and cultured in LB medium with kanamycin (50 mg L^−1^) and rifampind (25 mg L^−1^) antibiotics at 28 °C for 2 days (d). For the transient expression assays, *A. tumefaciens* strains were incubated in LB medium with appropriate antibiotics at 180 rpm overnight and then centrifuged and resuspended in infiltration buffer (10 mM MgCl_2_, 10 mM MES PH5.6 and 100 µM acetosyringone) to OD_600_ = 0.8. After 2 h incubation at room temperature in the dark, the resuspended *A*. *tumefaciens* carrying the target construct were infiltrated into 4- to 6-week-old *N*. *benthamiana* leaves using a 1 mL needless syringe. Target protein expression was assessed by Western blot after 2 d infiltration. The infiltrated leaves of *N. benthamiana* were ground in liquid nitrogen and suspended in 1 mL of lysis buffer (1 mM EDTA, 1% Triton X-100, 10 mM Tris-HCl, pH 8.0, 150 mM NaCl). The samples were incubated on ice for 10 min, followed by centrifugation at 12,000 rpm for 10 min at 4 °C to collect the supernatant containing soluble proteins. The extracted proteins were then mixed with 5× SDS-PAGE loading buffer (20315ES20, YEASEN, Shanghai, China) and denatured at 95 °C for 10 min. Subsequently, the denatured proteins were separated by SDS-PAGE and analyzed via Western blot using a FLAG M2 mouse monoclonal antibody (F3165, Sigma, St. Louis, MO, USA).

### 2.5. 3,3′-Diaminobenzidine (DAB) and Trypan Blue Straining

To detect cell death by trypan blue straining, the 5 d infiltrated leaves were treated with trypan blue solution at 95 °C for 10 min and then incubated at 37 °C overnight. The leaves were destained with chloral hydrate solution and photographed until the background was clear.

To observe the accumulation of ROS in the infiltrated *N*. *benthamiana* leaves, DAB straining was performed as previously described [8]. Briefly, *N*. *benthamiana* leaves were collected after 5 d of infiltration and placed in DAB solution (1 mg/mL, pH 3.8) for 30 min under vacuum at room temperature. The leaves were kept in the DAB solution for 6 h in the dark and then boiled in 75% (*v*/*v*) alcohol for a few minutes until the green color of the leaves had completely faded for proper visualization. Each experiment was performed at least three biological repeats.

### 2.6. Subcellular Localization of Fg28

For the subcellular localization of Fg28 in *N*. *benthamiana*, the coding sequence of Fg28 was inserted into the plasmid pCAMBIA1300-GFP via the BamHI and SalI restriction sites. The constructed pCAMBIA1300-Fg28-GFP and pCAMBIA1300-GFP were introduced into *Agrobacterium* strain GV3101 and transiently expressed in *N. benthamiana*. For plasmolysis, 10% NaCl was used. The fluorescence signal of the agroinfiltrated area was observed and photographed using an SP8 confocal microscope (Leica, Wetzlar, Germany) after 48 h and 72 h of infiltration.

### 2.7. Yeast Signal Sequence Trap System and Secretion Assay

For the yeast secretion assay, the Fg28SP sequence was cloned into the pSUC2 vector and transformed into the yeast strain YTK12 [33]. Transformant strains were then diluted and screened on CMD-W medium and YPRAA medium plates. YTK12 strains carrying empty pSUC2 vector or pSUC2-Avr1bSP were used as negative and positive controls, respectively. The invertase enzymatic activity was determined by observing the reduction of 2, 3, 5-triphenyltetrazolium chloride (TTC) to insoluble red-colored 1, 3, 5-triphenylformazan (TPF) [34,35]. Additionally, Fg28 coding sequence was inserted into the pKNTG-GFP plasmid together with its native promoter (1500 bp) by overlapping PCR. *F*. *graminearum* protoplasts were extracted from hyphae as previously described, with slight modifications [36]. Briefly, mycelia were harvested from the yeast extract-peptone-dextrose (YEPD) medium and enzymatic digested by driselase and lysing enzymes to isolate protoplasts. Then, 10 µg of the Fg28-pKNTG-GFP plasmid was transformed into 200 µL of protoplasts using the polyethylene glycol method [7,37]. The mycelial blocks of the Fg28-GFP strain and GFP control were used to infect the coleoptile of two-week-old wheat seedlings. The infected hyphae and GFP signals in wheat tissues were examined at 2 dpi.

### 2.8. Gene Knockout and Pathogenicity Testing

To generate *Fg28* gene deletion mutants, ~1 kb of upstream and downstream flanking sequences of *Fg28* were fused to the hygromycin phosphotransferase (*HPH*) fragments by double-jointed PCR [38]. The jointed PCR products were transferred into wild-type PH-1 strain protoplasts as previously described [7,37]. The mutants were selected with hygromycin B (50 µg mL^−1^) in PDA medium and subsequently verified by PCR. The primer sequences used for PCR are provided in Appendix A. DNA extraction was carried out using the Fastpure plant DNA isolation mini kit (DC104-01, Vazyme, China). Colony diameters and morphology of wild-type and mutant strains were measured and evaluated after 4 d of incubation at 28 °C in the dark.

For pathogenicity assays, WT and *Fg28* mutant conidia were harvested from the CMC medium cultured 5 d at 28 °C. The conidia were resuspended in sterile distilled water to OD_600_ = 1.0. Approximately twenty spikes were inoculated with 10 µL of conidial suspension and wrapped in plastic bags for 48 h. Disease spikelet number was investigated at 14 days post inoculation (dpi). In addition, to further evaluate the virulence of *Fg28* mutants, 2 µL of conidial suspension was inoculated onto coleoptiles of 4 d-old wheat seedlings as previously described [39]. Lesion sizes on the coleoptiles were measured at 5 dpi. The experiments were biologically repeated three times for each strain.

### 2.9. RNA Extraction and RT-qPCR Analysis

To characterize the *Fg28* expression during infection, wheat spikes infected with *F*. *graminearum* were collected at 24 h, 48 h and 72 h after inoculation for reverse transcription-quantitative polymerase chain reaction (RT-qPCR). *N*. *benthamiana* leaves were collected after 48 h infiltration for target gene expression detection. Total RNA was extracted using an RNAprep Pure Plant Kit (DP441, Tiangen Biotech, Beijing, China). Reverse transcription for cDNA synthesis was performed using the HiScript III 1st Strand cDNA synthesis kit (R312-02, Vazyme, China). ChamQ Universal SYBR qPCR Master Mix (Q711-02, Vazyme, China) was used for RT-qPCR analysis, and reactions were performed on a CFX Connect Real-Time System (Bio-Rad, Hercules, CA, USA). The specific PCR protocol was as follows: 95 °C for 3 min, followed by 39 cycles of 95 °C for 15 s and 60 °C for 1 min. The dissolution curve program was set from 65 °C to 90 °C in 0.5 °C increments, with each temperature lasting 5 s. The specificity of the PCR primers was verified using the dissolution curve. *FgTublin* and *Nbactin* were used as internal reference genes. The relative expression of each gene was calculated using the 2^−ΔΔCt^ method with three independent replications. The primers used for RT-qPCR are listed in Appendix A.

### 2.10. Statistical Analyses

All experiments were conducted at least three times. For RT-qPCR data, statistical analysis was performed for each experiment using Microsoft Excel 2019, and significance was determined using a *t*-test (*p* < 0.05). GraphPad Prism was used to generate the figures. For the assessment of *Fusarium graminearum* virulence, statistical analysis was conducted using GraphPad Prism version 8.0, with significance determined by a *t*-test (*p* < 0.05).

## 3. Results

### 3.1. Proteomic Analysis of the F. graminearum Secretome in the Apoplast Fluid of Wheat Coleoptiles

To identify the secretory proteins of *F. graminearum* in the apoplast fluid of wheat coleoptiles, the wheat coleoptiles were incubated with *F. graminearum* for 48 h, and the apoplast fluid was analyzed by liquid chromatography–tandem mass spectrometry (LC-MS/MS). A total of 79 candidate secretory proteins of *F. graminearum* were identified in the apoplast of wheat coleoptiles from three independent biological experiments (Appendix A, Figure 1A). A total of 16 of the 79 (20.3%) secretory proteins were predicted to have a secretory signal peptide (Appendix A). The heat map analysis using expression data from the previous study [27] shows that most of genes (48/79, ~64.9%) are highly expressed in plant-penetrating infection cushions (ICs) on the plant surface, while fewer are highly expressed in mycelia grown in complete medium (MY) and in epiphytic runner hyphae (RH) (Appendix A, Figure 1B). Moreover, 79 candidate secretory proteins were classified into 11 groups based on their predicted biological functions. Many of the secretory proteins were oxidoreductase (19%) and hydrolase (17.7%). The rest of the candidate secretory proteins consisted of transferase (12.7%), ribonucleoprotein and ribosomal protein (10.1%), other molecular function and cellular processes (8.9%), lyase (7.6%), heat shock/chaperone proteins (7.6%), isomerase (5.1%), ligase (1.3%), translocase (1.3%) and unknown function (8.9%) (Figure 1C). Furthermore, KEGG analysis shows that the 79 secretory proteins were enriched in biosynthesis of secondary metabolites, carbon metabolism and biosynthesis of amino acids (Figure 1D).

### 3.2. Fg28 Induces Cell Death in N. benthamiana

To further investigate the potential function of these secretory proteins of *F. graminearum*, the coding sequences of 16 proteins with secretory signal peptides were cloned and transiently expressed in *N*. *benthamiana* leaves. Green fluorescent protein (GFP) was used as a negative control. A secretory metalloprotease protein Fg28 (encoded by *FGSG_00028*) containing a Peptidase_M43 domain and a signal peptide was identified that induced cell death in *N*. *benthamiana* after 5 d of infiltration (Figure 2A,B). In comparison with the GFP, macroscopic cell death was observed in the zone with Fg28 expression by trypan blue staining (Figure 2C). Furthermore, DAB staining showed that Fg28 induced significant ROS accumulation compared to the GFP control (Figure 2D). The expression of Fg28 in infiltrated leaves was confirmed by Western blot (Figure 2E, Appendix A). Thus, we identified a cell-death-inducing protein of *F. graminearum* from the apoplast fluid of wheat coleoptiles.

### 3.3. Fg28 Is Highly Up-Regulated and Secreted During the Early Stages of F. graminearum Infection

To further verify the features of Fg28, we first analyzed the expression level of *Fg28* during the early stage of *F*. *graminearum* infection by reverse transcription-quantitative polymerase chain reaction (RT-qPCR). Wheat spikes of Jimai22 were inoculated with fresh *F*. *graminearum* conidia and collected at four time points including 0 h, 24 h, 48 h and 72 h post inoculation (hpi). The RT-qPCR results show that *Fg28* is obviously up-regulated by 6- to 9-fold and peaking at 48 hpi during the early stage of *F. graminearum* infection (Figure 3A).

The secretory function of the Fg28 signal peptide was tested using the yeast signal trap assay. Like the positive control pSUC2-Avr1bSP, the yeast transformants containing pSUC2-Fg28SP grew well on selective YPRAA medium plate, but YTK12 and the strain carrying the pSUC2 vector used as a negative control failed to grow on the YPRAA plate (Figure 3B). The enzyme activity of the secreted invertase was detected by the TTC assay, and the secreted invertase of the transformants containing pSUC2-Avr1bSP or pSUC2-Fg28SP converted TTC into the insoluble red-colored TPF, but YTK12 and the strain carrying the pSUC2 vector did not (Figure 3B). These results support the functionality of the Fg28 signal peptide. Given the signal peptide sequence in the N-terminal region of Fg28, we next tested whether the Fg28 could be secreted by *F*. *graminearum* into the wheat apoplast. In the Fg28-GFP transgenic strain, GFP fluorescence was observed through the mycelial cells of *F*. *graminearum* (Figure 3C). The Fg28-GFP and free GFP overexpression strains were then inoculated onto wheat coleoptiles. After 2 dpi, the fluorescence signal of the Fg28-GFP strain was observed in the wheat intercellular space, but no signals of the GFP overexpression strain were observed in the wheat cells or apoplast (Figure 3D). Taken together, the results indicate that Fg28 is up-regulated and secreted into the intercellular space during the early stage of *Fusarium* infection.

### 3.4. Full Length of Fg28 Required to Induce Cell Death

To determine which region of Fg28 is necessary for inducing the cell death, a series of mutant recombinant Flag tagged proteins of Fg28 (Fg28^ΔM43^, Fg28^ΔSP^, Fg28^SP+M43^, Fg28^ΔC200^ and Fg28^ΔC400^) were constructed (Figure 4A) and transiently expressed in *N*. *benthamiana* leaves. The results showed that the full length of Fg28 induced cell death, whereas all the deletion mutants lost the ability to induce cell death. ROS staining and trypan blue also confirmed this result (Figure 4B,C). Western blot analysis confirmed the expression of full-length Fg28 protein and Fg28 truncated proteins (Figure 4D and Appendix A).

In addition, a GFP fusion of the full-length Fg28 protein was constructed and transiently expressed in *N*. *benthamiana* leaves. The results show that Fg28 is located at the plasma membrane (Figure 5A). NaCl-induced plasmolysis further demonstrated the subcellular location of Fg28 at the plasma membrane of *N*. *benthamiana* (Figure 5B). Interestingly, when Fg28 was expressed for 72 h, Fg28-GFP formed bright fluorescent aggregates near the plasma membrane, suggesting that Fg28 may have a specific function at the plasma membrane. Taken together, these results indicate that the full length of Fg28 is required for its function in inducing cell death and that the cell death induced by Fg28 may be related to its distinctive subcellular location.

### 3.5. Fg28 Induces Plant Immune Response and Is Independent of BAK1/SOBIR1 and EDS1/PAD4

To further investigate the mechanisms of Fg28-induced cell death, we analyzed the expression levels of nine typically immune response genes [6,12,15,40]. The RT-qPCR results showed that all five typical PTI marker genes, *NbWRKY7*, *NbWRKY8*, *NbACRE31*, *NbPIA5* and *NbPTI5*, and one hypersensitivity gene, *NbHIN1*, were significantly up-regulated in the Fg28 Flag region compared to the GFP Flag region after 48 h of transient expression in *N*. *benthamiana* (Figure 6A). This result suggests that Fg28 induces a significant PTI immune response.

To test whether Fg28-induced immune responses and cell death are dependent on the classical BAK1/SOBIR1 and EDS1/PAD4 immune pathways, Fg28-GFP fusion proteins were transiently expressed in the *NbBAK1*, *NbSOBIR1*, *NbEDS1* and *NbPAD4* knockout mutants of *N*. *benthamiana*. The results showed that Fg28 caused ROS accumulation and cell death in all four mutants, suggesting that Fg28-induced immunity is not dependent on these four classical immune pathway proteins. The results were further confirmed by DAB staining and trypan blue (Figure 6B–E and Appendix A). Thus, our results suggest that Fg28 induces cell death and immune responses that are independent of BAK1/SOBIR1 and EDS1/PAD4.

### 3.6. Knockout Fg28 Have no Defect on the Virulence of F. graminearum

To investigate the role of *Fg28* in *F*. *graminearum* infection, the *Fg28* gene was replaced by the *proTrpc::HPH* cassette to obtain *Fg28* mutants (Appendix A). The colony morphology and the mycelial growth rate of the *Fg28* mutant strains were similar to wild-type PH-1 stains (Figure 7A,D). The number of diseased spikelets was calculated to evaluate the virulence of WT and Δ*Fg28* strains by inoculating with wheat cultivar Fielder. The number of diseased spikelets was similar between WT- and Δ*Fg28*-inoculated wheat (Figure 7B,E). In addition, the virulence of *Fg28* mutant strains on wheat coleoptiles was also investigated. The results show that *Fg28* mutants do not influence the virulence of *F*. *graminearum* on wheat coleoptiles (Figure 7C,F). These results suggest that *Fg28* does not affect the virulence of *F*. *graminearum.*

## 4. Discussion

Various secretory proteins facilitate pathogen colonization and play crucial roles in pathogen–plant interaction [21,40,41,42,43,44]. To identify the secreted proteins of *F. graminearum*, several studies have been carried using different methods. A total of 289 proteins (229 in vitro and 120 in planta) were identified as potential secretory proteins by MS/MS in 13 media in vitro and in planta [45]. In addition, 88 putative effectors were identified based on the transcriptomic analyses of runner hyphae (RH) and infection cushions (IC) [27]. Recently, the proteome of the *F. graminearum* secretome in potato dextrose broth medium was identified in two studies, and the Fg12, Fg62 and five FgSLP proteins were found to be secreted and contribute to *F. graminearum* virulence [6,8,12]. The apoplast is the primary space of the interaction between pathogens and plants. However, no data are available on the secretory proteins of *F. graminearum* in the plant apoplast. This study provides the first secretory proteome dataset of *F. graminearum* in the apoplast of wheat coleoptiles. A total of 79 potential secretory proteins were identified, of which 16 had signal peptides, and most of the proteins (48/79, ~64.9%) were expressed in IC, indicating the reliability of this dataset. Similar to previous studies, 64.6% of these 79 proteins were annotated to different enzymes (Appendix A, Figure 1A–C) [6,21]. These results also demonstrate that *F. graminearum*, as a hemibiotrophic fungus, may obtain nutrients mainly through enzymatic digestion of host components as previous study [6]. In conclusion, this study provides a set of data on *F. graminearum* secretory proteins in the wheat apoplast, which will contribute to understand the interaction between wheat and *F. graminearum*.

The immune response elicited by pathogen-secreted proteins may involve multiple mechanisms. Classical immunity concept suggests that plants activate the PTI and ETI immune responses by recognizing PAMPs and effectors of pathogens, respectively. BAK1 is a key molecule that interacts with PRRs to form a complex, playing a crucial role in plant PTI [46]. Similarly, SOBIR1 is a well-studied receptor-like kinase that contributes to plant immunity and PAMP recognition [47,48]. In ETI, EDS1 interacts with PAD4 to form heterodimers that facilitate TNL-mediated immunity [49]. Plants can also directly recognize extracellular pathogen-secreted proteins to initiate PTI. For example, the extracellular secreted protein PsAPE1 was recognized by BAK1 and triggered the PTI immune response [21]. At the same time, plants can also recognize DAMPs generated by pathogen-induced damage and trigger an immune response. For example, the secreted proteins MoCel12A and MoCel12B from *Magnaporthe oryzae* degrade hemicellulose to produce oligosaccharides that act as DAMPs to activate plant immunity via OsCERK1 during *M. oryzae* infection in rice [50]. However, there are instances where certain pathogen-secreted proteins elicit robust immune responses and cell death, independent of the core PTI and ETI pathways. For example, the five FgSLP toxic proteins induce cell death by protein degradation, independent of the PTI and ETI pathways [8]. The effector recognition receptor RXEG1 specifically recognizes PsXEG1 and activates downstream immune responses [19]. In this study, we identified a secreted metalloprotease, Fg28, which induces cell death and reactive oxygen species accumulation. Similarly with previous studies, *Fg28* was highly expressed during the early stages of infection (Figure 3A), suggesting that it may be involved in the process of destruction cellular components and invasion of plant tissues during this period. In addition, significant up-regulation of PTI and HR marker genes was observed in the plant response to Fg28, suggesting that Fg28 induces immune responses (Figure 6A). However, Fg28-induced cell death and ROS accumulation did not depend on the core PTI and ETI pathways, suggesting that Fg28 induces immune responses independently of typical immune response pathways (Figure 6B–E).

The signal peptides of secreted proteins and their functional structural domains play a crucial role in determining their functionality. For example, the removal of the signal peptide and the mutation of the enzyme site in Fg12 both failed to induce cell death [6]. Similarly, in this study, the removal of the signal peptide and the disruption of the enzyme structure of Fg28 resulted in the loss of the ability of Fg28 to induce cell death, indicating that both the signal peptide and the enzyme activity of Fg28 are essential for its function. Furthermore, the region linking the signal peptide to the structural domain of the enzyme is also essential, and its deletion may have disrupted the Fg28 protein structure, thereby failing to induce cell death. The full length of the Fg28 protein is, therefore, essential for its function (Figure 4). Secreted enzyme proteins have been shown to disrupt and degrade host biological components, such as APE1 and XEG1 [21,51]. The localization of the full-length Fg28-GFP to the cell membrane was abnormal, and the protein aggregated after 48 h of expression in *N. benthamiana* (Figure 5). This suggests that the protein may disrupt the structure of the cell membrane, leading to cell death and immune response. In conclusion, our results have identified a protein with the potential to cause cell membranes toxicity and induce an immune response.

Pathogen secretory proteins play an important role in pathogen virulence and have been identified in a number of pathogens, particularly that proteins induce cell death. For example, the secreted aldose 1-epimerase (AEP1) in *P. sojae* induces cell death in *N. benthamiana* and acts as a virulence factor [21]. In *F. graminearum*, several secreted proteins, including the RNase Fg12, the effector Fg62, and the subtilisin-like proteases FgSLPs, also induce plant cell death and contribute to the virulence of *F. graminearum* [6,8,12]. Exceptionally, some secretory proteins may have little or no effect on pathogenicity. The absence of Zt6 in *Zymoseptoria tritici* did not affect its pathogenicity. However, Zt6 may play an important role in protecting the pathogen from other microorganisms and defending its ecological niche, rather than directly causing host damage [52]. In this study, similar to the slow cell death phenotype, the knockout of *Fg28* had no effect on virulence, suggesting that Fg28 may be a less toxicity protein (Figure 7). However, this does not mean that low toxicity proteins are not important for the pathogenicity of pathogens. There may be an evolutionary pattern of competition between *Fusarium* and wheat. The pathogen secretes a number of weak pathogenic proteins when it infects the plant, but the loss of one of these proteins does not affect its pathogenicity and thus favors the infection and survival of *F. graminearum*. In correspondence, the evolutionary analysis of metalloproteases in seven *Fusarium* genus strains and five other crop fungi pathogens, as well as four reported plant pathogenic metalloproteases (one M43-class and three M36-class), also supports that M43-class metalloproteases in *F. graminearum* have formed a distinct phylogenetic clade. This suggests that these proteins have undergone further species-specific diversification within *F. graminearum* (Appendix A). Such diversification may be associated with their relatively lower pathogenicity. Therefore, we propose that the pathogen secretes numerous weakly pathogenic proteins during plant infection; however, the loss of any one of these proteins does not significantly affect its overall pathogenicity, thereby enhancing the infection and survival capabilities of *F. graminearum*.

## 5. Conclusions

Conclusively, the present study reports the first *F. graminearum* secretory proteome in apoplast of wheat and identifies a protein (Fg28) that elicits immune response and cell death. However, the mechanism responsible for the cell death elicitation remains to be further explored. The present study will provide new insights into the interactions between *F. graminearum* and wheat.

## Figures and Tables

**Figure 1 jof-11-00240-f001:**
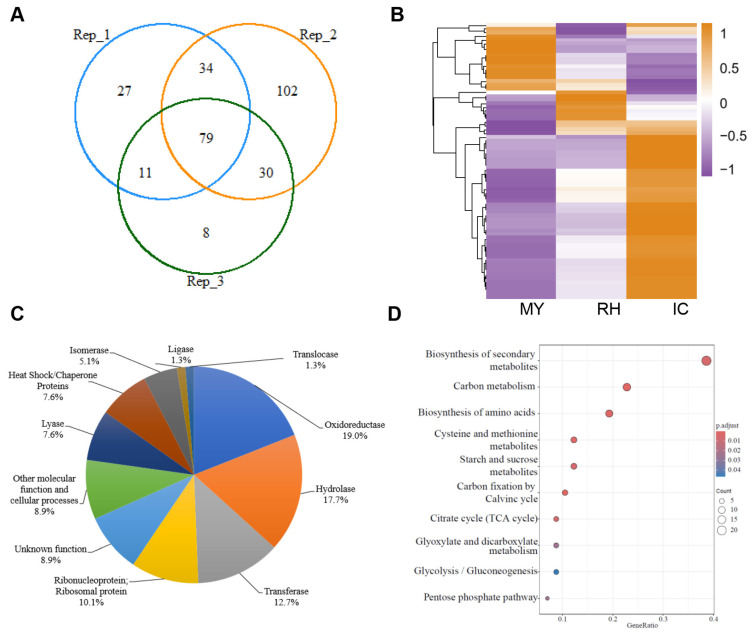
The secretory proteome in the apoplast of wheat coleoptiles during the early stage of *Fusarium graminearum* infection. (**A**) Venn diagram showing the overlap of *F. graminearum* secretory proteins in apoplast identified by liquid chromatography–tandem mass spectrometry (LC-MS/MS) in three biological replicates; (**B**) gene expression heat map of 79 overlapping secretory proteins. Expression levels were normalized in all three types, including mycelia grown in complete medium, epiphytically growing runner hyphae (RH) and plant-penetrating infection cushions (IC) on the plant surface; (**C**) categories of the identified secretory proteins. Secretory proteins were classified into 11 different groups based on their biological functions; (**D**) KEGG pathway enrichment analysis of the overlapping 79 secretory proteins of *F. graminearum*.

**Figure 2 jof-11-00240-f002:**
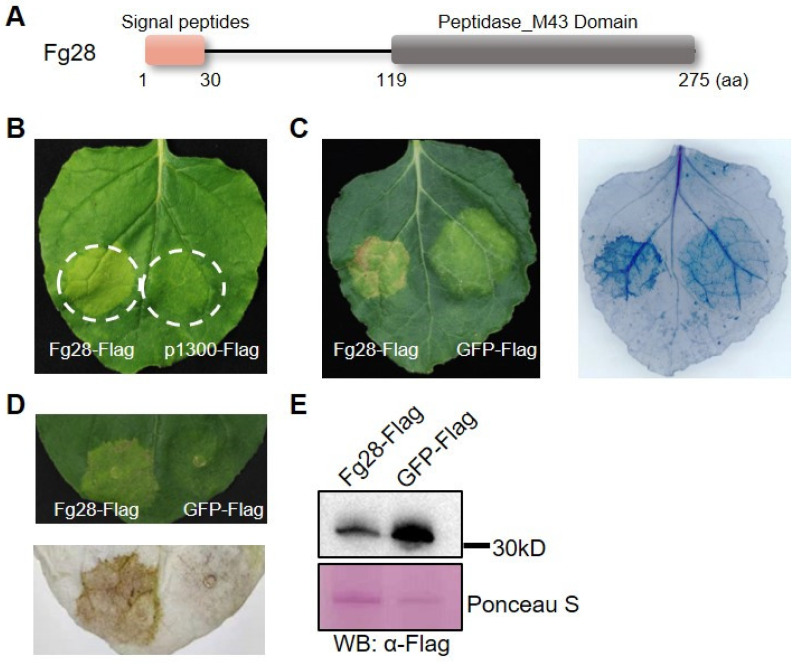
*Fg28* induces ROS accumulation and cell death in *Nicotiana benthamiana*. (**A**) Schematic representation of the *Fg28* protein domains; (**B**) transient expression of *Fg28* induces cell death in *N. benthamiana*. Representative images were taken at 5 days post inoculation (dpi); (**C**) *Fg28*-induced cell death on *N. benthamiana* leaves was stained by trypan blue solution; (**D**) *Fg28*-induced reactive oxygen species on *N. benthamiana* leaves were stained with 3,3′-diaminobenzidine (DAB); (**E**) *Fg28*-Flag and GFP-Flag protein levels in *N. benthamiana* leaves were detected by Western blot using anti-Flag antibody. Ponceau staining of Rubisco was used as a loading control.

**Figure 3 jof-11-00240-f003:**
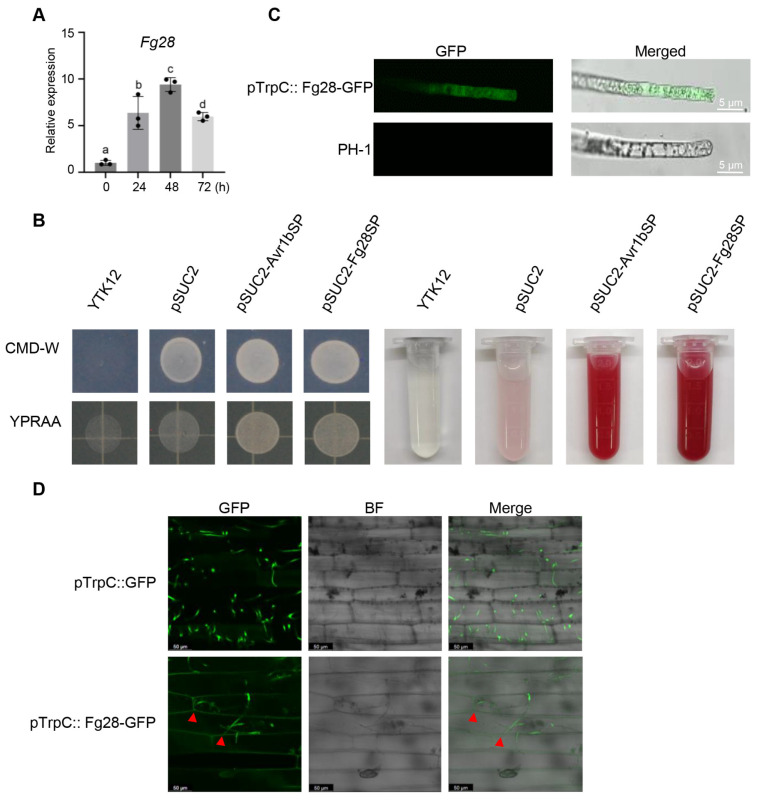
*Fg28* secreted into wheat apoplast during the early stages of *F. graminearum* infection. (**A**) Relative transcript levels of *Fg28* during different stages of infection in wheat. Significant differences are indicated by different letters; (**B**) functional validation of the *Fg28* signal peptides using yeast invertase secretion assay. Yeast transformants were cultured on CMD-W and YPRAA plates. The enzymatic activity of the secreted invertase was detected by reduction of 2, 3, 5-triphenyltetrazolium chloride (TTC) to insoluble red-colored 1, 3, 5-triphenylformazan (TPF); (**C**) the subcellular localization of the *Fg28* in *F. graminearum* hyphal cells; (**D**) the *Fg28*-GFP fluorescence was detected in wheat cells. Fluorescence was observed and measured 2 dpi in wheat coleoptiles. The triangles indicated the secretion of GFP fluorescence in wheat cells.

**Figure 4 jof-11-00240-f004:**
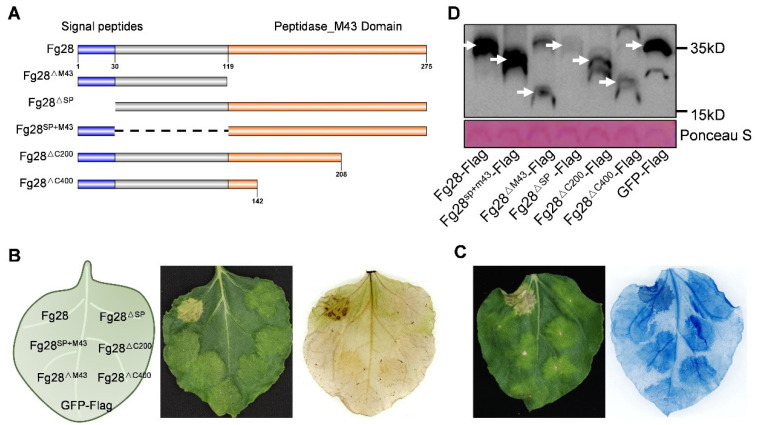
The full-length protein is essential for *Fg28* function. (**A**) Schematic representation of *Fg28* and its deletion mutants (*Fg28*^Δ43^, *Fg28*^ΔSP^, *Fg28*^Fg28SP+M43^, *Fg28*
^ΔC200^ and *Fg28*^ΔC400^); (**B**,**C**) *Fg28* and its deletion mutants induced cell death at 5 dpi in *N*. *benthamiana*. ROS was determined by DAB staining and cell death were stained with trypan blue solution. (**D**) *Fg28*-Flag and its deletion mutants-flag proteins was detected by Western blot using the anti-Flag antibody. Ponceau staining of Rubisco was used as a loading control. White arrows indicate the target proteins.

**Figure 5 jof-11-00240-f005:**
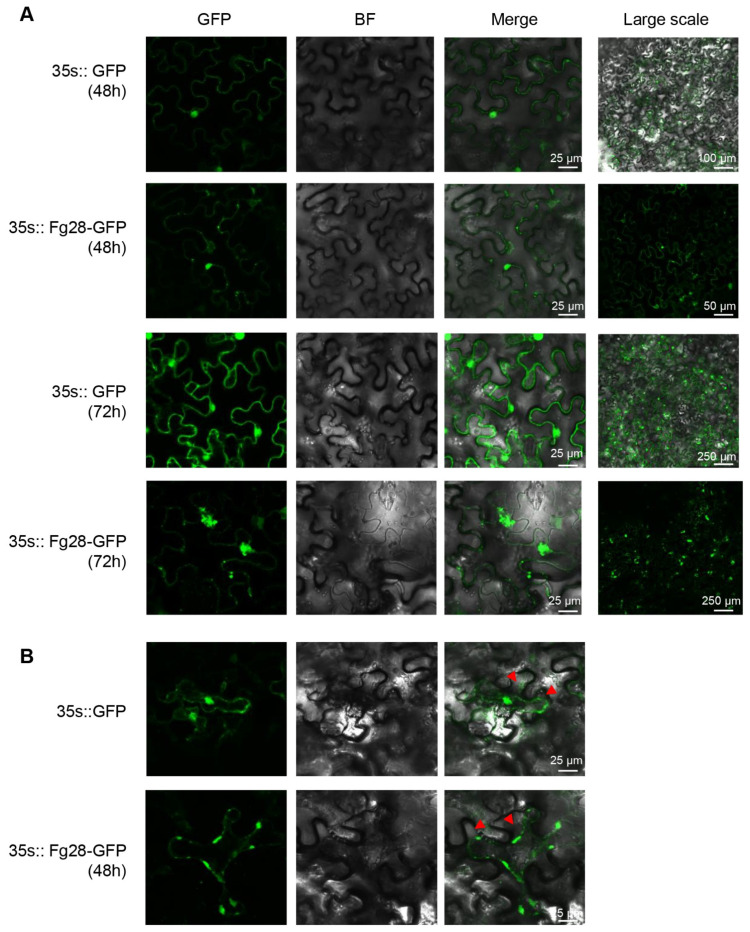
*Fg28* function on the cell membrane. (**A**) *Fg28*-GFP and free GFP control were expressed and photographed at 48 h and 72 h after agroinfiltration in *N. benthamiana* leaves. A significant increase in green fluorescence aggregates was observed in the vicinity of the cell membrane after 72 h of *Fg28*-GFP expression. (**B**) The presence of *Fg28* on the cell membrane was confirmed through plasmolysis. Plasmolysis was performed by 10% NaCl treatment. Triangles indicate the apoplast.

**Figure 6 jof-11-00240-f006:**
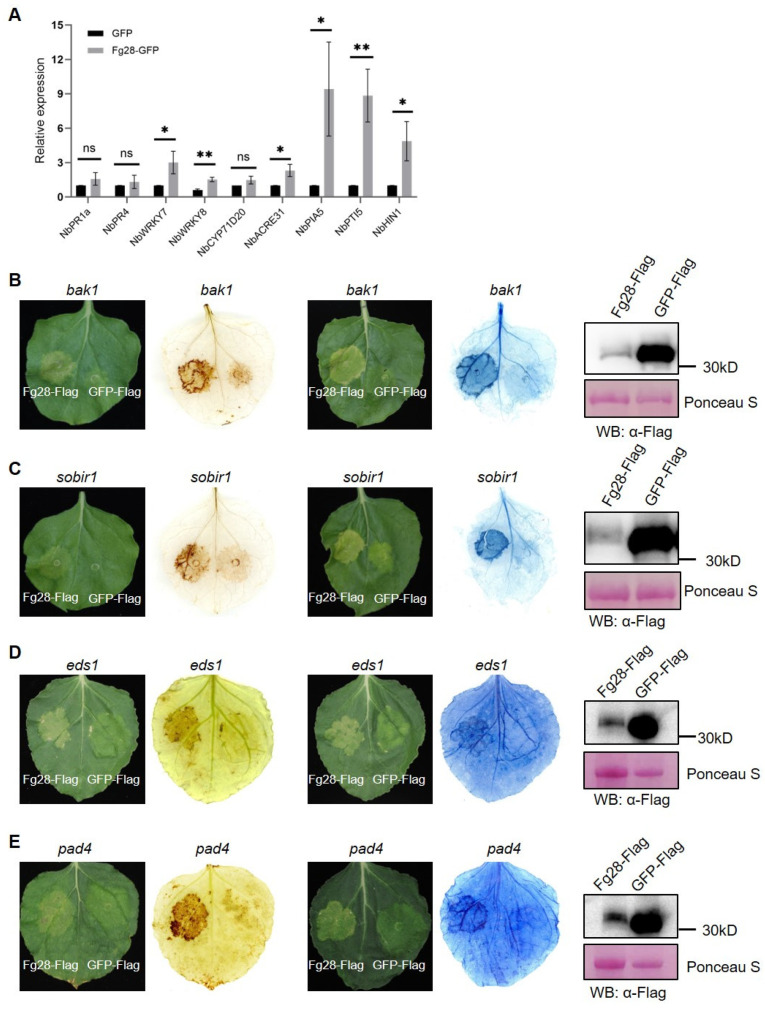
*Fg28* induces immune response in *N. benthamiana* leaves but is independent of BAK1, SOBIR1, EDS1 and PAD4. (**A**) Relative transcription levels of plant defense-related genes (*NbPR1* and *NbPR4*), PTI marker genes (*NbWRKY7*, *NbWRKY8*, *NbCYP71D20*, *NbACRE31*, *NbPIA5* and *NbPTI5*) and a hypersensitive response gene (*NbHIN1*) in *N*. *benthamiana*. *NbActin* was used as an internal reference gene. Asterisks indicate significant differences based on multiple *t* test (**, *p* < 0.01; *, *p* < 0.05; ns, *p* > 0.05); (**B**–**E**) BAK1 (**B**), SOBIR1 (**C**), EDS1 (**D**) and PAD4 (**E**) are not required for *Fg28*-induced cell death in *N*. *benthamiana.* ROS was assessed by DAB staining at 5 dpi in *N. benthamiana*. Cell death was assessed by trypan blue staining. *Fg28*-Flag and GFP-Flag were detected by Western blot using the anti-Flag antibody. Ponceau staining of Rubisco was used as a loading control.

**Figure 7 jof-11-00240-f007:**
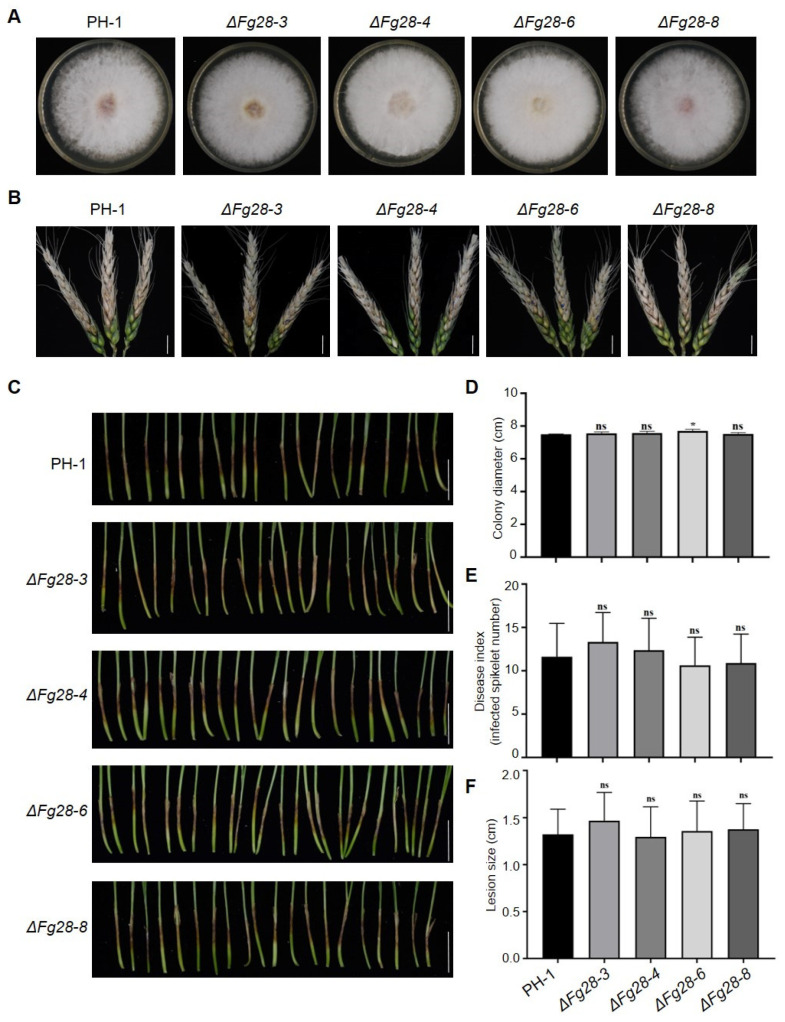
*Fg28* does not affect the virulence of *F. graminearum*. (**A**,**D**) Colony morphology and diameter of the wild-type (PH-1) and *Fg28* knockout mutant strains (Δ*Fg28-3*, Δ*Fg28-4*, Δ*Fg28-6* and Δ*Fg28-8*) cultured on potato dextrose agar at 28 °C for 4 d. The mean and standard deviation of colony diameter were calculated from three independent experiments (*, *p* < 0.05; ns, *p* > 0.05); (**B**,**E**) the disease index of wheat heads inoculated with PH-1 or *Fg28* -knockout mutant strains (Δ*Fg28-3*, Δ*Fg28-4*, Δ*Fg28-6* and Δ*Fg28-8*). The mean and SD were obtained from three independent replicates, each containing at least 20 spikes. Asterisks indicate a significant difference based on unpaired two-tailed Student’s *t*-test (*, *p* < 0.05; ns, *p* > 0.05). Scale bars, 1 cm. (**C**,**F**) Lesion sizes on wheat coleoptiles inoculated with PH-1 or *Fg28* knockout (Δ*Fg28-3*, Δ*Fg28-4*, Δ*Fg28-6* and Δ*Fg28-8*) mutant strains. Mean lesion sizes were calculated from three independent replicates, each containing at least 15 seedlings. Asterisks indicate a significant difference based on unpaired two-tailed Student’s *t*-test (*, *p* < 0.05; ns, *p* > 0.05). Scale bars, 1 cm.

## Data Availability

The original contributions presented in this study are included in the article/Appendix A. Further inquiries can be directed to the corresponding authors.

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
