# Peer review of "Proteomic Analysis of the Fusarium graminearum Secretory Proteins in Wheat Apoplast Reveals a Cell-Death-Inducing M43 Peptidase"

_jof, 2025, doi:10.3390/jof11040240_

Round 1
Reviewer 1 Report
The manuscript addresses a relevant subject of investigation but the experimental design does not allow me to assess the validity of results and experimental rigor.
The methodology used to perform Western blotting assays is not included. Control Lines with negative/positive controls are missed. The authors must include the unedited membrane images for inspection.
The authors did not include any strategy to assess ectopic integrations when generating the null mutant. Consequently, there is no direct link between the phenotype and the disrupted gene.
Please include references for the use of those reference genes in the RT-qPCR experiments.
Please include a section for statistical analyses applied to data. Describe which tests were used for each experiment.
Please include methodological details for the mass spectrometry analysis.
The images of vegetal tissues lack a scale bar.
Figure 7.- What are all the strains shown in this figure?
The manuscript addresses a relevant subject of investigation but the experimental design does not allow me to assess the validity of results and experimental rigor.
The methodology used to perform Western blotting assays is not included. Control Lines with negative/positive controls are missed. The authors must include the unedited membrane images for inspection.
The authors did not include any strategy to assess ectopic integrations when generating the null mutant. Consequently, there is no direct link between the phenotype and the disrupted gene.
Please include references for the use of those reference genes in the RT-qPCR experiments.
Please include a section for statistical analyses applied to data. Describe which tests were used for each experiment.
Please include methodological details for the mass spectrometry analysis.
The images of vegetal tissues lack a scale bar.
Figure 7.- What are all the strains shown in this figure?
Author Response
Dear Editor and Reviewers,
We greatly appreciate your constructive comments and suggestions that have helped us to significantly improve the manuscript for this revision. We have now made major revisions of our manuscript according to the reviewers' suggestions. We made point-to-point response as below with our responses highlighted in Blue. The revised manuscript text shows major changes highlighted in red.
Reviewer 1
The manuscript addresses a relevant subject of investigation but the experimental design does not allow me to assess the validity of results and experimental rigor.
Response: We appreciate the reviewer's time and effort, and thank the reviewer for helping us to improve our presentation.
- The methodology used to perform Western blotting assays is not included. Control Lines with negative/positive controls are missed. The authors must include the unedited membrane images for inspection.
Response: Thank you for highlighting the lack of detailed methodology for the Western blotting assays. We have included a comprehensive description in the revised manuscript under Section 2.4 (Materials and Methods) from lines 138 to 146.
Since the antibodies and experimental conditions remained consistent across our Western blot analyses, negative and positive controls were not included in every experiment. As illustrated in Fig. I, the Western blot image demonstrates the specificity and reliability of the antibody used in the assay: the first lane represents the negative control (tobacco leaves injected with infiltration buffer only), while the second lane corresponds to the positive control (GFP-Flag). This configuration validates the antibody's performance under the established experimental conditions.
Fig. I. The expression levels of GFP-Flag protein in N. benthamiana leaves were analyzed by Western blot using an anti-Flag antibody, with Rubisco protein detected by Ponceau S staining serving as the loading control.
As suggested, we have included the unedited membrane images for review and inspection (see Fig. II below).
Fig. II. The original image of Western blot in this paper. A, Original image of Figure 2E; B, Original image of Figure 4D; C, Original image of Figure 6A and B; D, Original image of Figure 6D and E.
- The authors did not include any strategy to assess ectopic integrations when generating the null mutant. Consequently, there is no direct link between the phenotype and the disrupted gene.
Response: We sincerely appreciate the reviewer's valuable comment. We conducted phenotypic characterization using four independent Fg28 knockout mutant strains (ΔFg28-3, ΔFg28-4, ΔFg28-6, and ΔFg28-8), all of which exhibited consistent phenotypes. This observation eliminates the potential influence of ectopic integration on the mutant phenotypes.
- Please include references for the use of those reference genes in the RT-qPCR experiments.
Response: As suggested, these references (6, 12, 15, 40) have been added to the revised manuscript (Results section 3.5, line 342).
- Please include a section for statistical analyses applied to data. Describe which tests were used for each experiment.
Response: Thank you for your valuable comment regarding the statistical analyses in our study. We have now included a detailed description of the statistical analyses applied to our data. The specific tests used for each experiment are described in the revised manuscript under the Materials and Methods section 3.0, line 217-222.
- Please include methodological details for the mass spectrometry analysis.
Response: As suggested, we have now included the methodological details for the mass spectrometry analysis in the revised manuscript (Materials and methods section 2.2, line 101-110).
- The images of vegetal tissues lack a scale bar.
Response: We have now included the scale bar in the Figure 7B and 7C.
- Figure 7.- What are all the strains shown in this figure?
Response: Thank you for your question. Figure 7 includes the following strains: the wild-type (PH-1) and four Fg28 knockout mutant strains (ΔFg28-3, ΔFg28-4, ΔFg28-6, and ΔFg28-8). To ensure clarity and transparency, we have updated the figure legend in the revised manuscript to explicitly state this information.
Reviewer 2 Report
The manuscript “Proteomic analysis of the Fusarium graminearum secretory proteins in wheat apoplast reveals a cell death inducing M43 peptidase” by Li and co-authors analyzed Fusarium graminearum secretory proteins in wheat apoplast and identified a metalloprotease up-regulated in the early stages of infection and able to induce cell death and reactive oxygen species (ROS) accumulation in Nicotiana benthamiana. The work and the results obtained are clearly presented, the analyses conducted are exhaustive and carried out appropriately.
I suppose Fielder and Jimai22 wheat cultivars that the authors have used for artificial infection, are susceptible to F. graminearum. Have the authors any evidence of Fg28 protein production in the apoplast of resistant wheat cultivars? It might be useful to test it in the frame of defining Fg28 role.
Line 62: add “and” before “depends”
Line 97: replace OD with OD600
Line 103: add “with” before “three”
Line 119: replace rifampind with rifampicin
Line 128 and 133: replace “straining” with “staining”
Line 138: add “with” before “at least”
Line 155: replace F. with Fusarium
Line 184: replace N. with Nicotiana
Line 221: add “(MY)” after “medium”
Line 347: replace “with” with “on”
Fig 7D: From this graph I don’t understand why DFg28-6 is significantly different (*) from the others
Line 400-401: please cite these “previous studies”
Line 402: add “of” before “cellular”
Line 407: add “from” before “typical”
Line 426: replace “that” with “those” and “induce” with “inducing”
Author Response
Dear Editor and Reviewers,
We greatly appreciate your constructive comments and suggestions that have helped us to significantly improve the manuscript for this revision. We have now made major revisions of our manuscript according to the reviewers' suggestions. We made point-to-point response as below with our responses highlighted in Blue. The revised manuscript text shows major changes highlighted in red.
Reviewer 2
The manuscript “Proteomic analysis of the Fusarium graminearum secretory proteins in wheat apoplast reveals a cell death inducing M43 peptidase” by Li and co-authors analyzed Fusarium graminearum secretory proteins in wheat apoplast and identified a metalloprotease up-regulated in the early stages of infection and able to induce cell death and reactive oxygen species (ROS) accumulation in Nicotiana benthamiana. The work and the results obtained are clearly presented, the analyses conducted are exhaustive and carried out appropriately.
Response: We thank the reviewer for your positive comments and insightful suggestions.
- I suppose Fielder and Jimai22 wheat cultivars that the authors have used for artificial infection, are susceptible to F. graminearum. Have the authors any evidence of Fg28 protein production in the apoplast of resistant wheat cultivars? It might be useful to test it in the frame of defining Fg28 role.
Response: We appreciate the reviewer's insightful comments. To address this point, we performed a comparative analysis of Fg28 gene expression patterns in both susceptible (Fielder) and resistant (Yangmai16 and Sumai 3) wheat cultivars at 48 hours post-inoculation. Quantitative RT-PCR analysis revealed comparable expression levels of Fg28 across these three cultivars with distinct Fusarium head blight resistance backgrounds (Fig. III). These findings support our hypothesis that the secretion of Fg28 protein may be independent of the host's resistance levels. We fully acknowledge that examining the presence of Fg28 in the apoplast could provide further clarity on its functional role in the resistance mechanism. This is an excellent suggestion, and we will certainly incorporate this aspect into our future research plans to gain a more comprehensive understanding of the interaction between Fg28 and wheat resistance.
Fig. III. Relative transcript levels of Fg28 after 48 h of infection in wheat FLD (Fielder), Yangmai 16 and Sumai 3.
Reviewer 3 Report
This very interesting paper is devoted to the analysis of the set of secretory proteins produced during the F.graminearum infection in wheat apoplast. Indeed, despite the fact that Fusarium is among the most devastating pathogens of cereals, the profile of proteins responsible for plant-pathogen interactions, including secretory proteins, is poorly discovered. This paper contains interesting and important information, the results are described compehensively and supported by the methods used. I believe the paper can be accepted for publication in JoF. At the same time, I have some optional notes, that would improve the text quality and significance.
- The authors mentioned such factors of plant immune response, as BAK1, SOBIR1, EDS1 and PAD4 and declared that Fg28-induced response was independent of them. I believe there should be more information and corresponding references regarding these proteins and their functional roles in plant immune reactions (In Introsuction and/or Discussion).
- The authors should provide information about conditions/profiles of PCR reactions, including those used to amplify F. graminearum cDNA and for RT-qPCR analysis. At least, annealing temperatures should be provided in Table S1. Also, were these primers developed in this work or taken form the literature?
- Are there any Fg28 homologs in other Fusarium/fungal/eukaryotic organisms and is there any information on their functional roles? If yes, the authors could provide a phylogenetic tree that would include Fg28 and its nearest homologs to clarify evolutionary relationships between them (OPTIONAL).
-
Author Response
Dear Editor and Reviewers,
We greatly appreciate your constructive comments and suggestions that have helped us to significantly improve the manuscript for this revision. We have now made major revisions of our manuscript according to the reviewers' suggestions. We made point-to-point response as below with our responses highlighted in Blue. The revised manuscript text shows major changes highlighted in red.
Reviewer 3
This very interesting paper is devoted to the analysis of the set of secretory proteins produced during the F.graminearum infection in wheat apoplast. Indeed, despite the fact that Fusarium is among the most devastating pathogens of cereals, the profile of proteins responsible for plant-pathogen interactions, including secretory proteins, is poorly discovered. This paper contains interesting and important information, the results are described compehensively and supported by the methods used. I believe the paper can be accepted for publication in JoF. At the same time, I have some optional notes, that would improve the text quality and significance.
Response: We thank the reviewer for your positive comments and insightful suggestions.
- The authors mentioned such factors of plant immune response, as BAK1, SOBIR1, EDS1 and PAD4 and declared that Fg28-induced response was independent of them. I believe there should be more information and corresponding references regarding these proteins and their functional roles in plant immune reactions (In Introsuction and/or Discussion).
Response: Thank you for your valuable suggestion. We have revised the Discussion section (lines 417-422) to include additional details on these key immune factors, along with appropriate references (46-50) to support their functional roles in plant immunity.
- The authors should provide information about conditions/profiles of PCR reactions, including those used to amplify F. graminearum cDNA and for RT-qPCR analysis. At least, annealing temperatures should be provided in Table S1. Also, were these primers developed in this work or taken form the literature?
Response: We have now included the detailed profiles for RT-qPCR analysis in the revised manuscript (Materials and methods section 2.9, line 210-213). Additionally, we have now included annealing temperatures in Table S1. Furthermore, in the article, the primers related to Fg28 were designed by ourselves, while the RT-qPCR analysis of typical immune response gene transcript levels in N. benthamiana was conducted following the methodology from the cited reference in Results section 3.5.
- Are there any Fg28 homologs in other Fusarium/fungal/eukaryotic organisms and is there any information on their functional roles? If yes, the authors could provide a phylogenetic tree that would include Fg28 and its nearest homologs to clarify evolutionary relationships between them (OPTIONAL).
Response: Thanks for your professional suggestion. We conducted a blastP analysis on 7 other fungi (Fusarium pseudograminearum, Fusarium proliferatum, Fusarium sporotrichioides, Fusarium poae, Fusarium solani, Fusarium oxysporum and Fusarium verticillioide) from the Fusarium genus and 5 important crop pathogens (Ustilago maydis, Ustilaginoidea virens, Zymoseptoria tritici, Magnaporthe oryzae and Verticillium dahliae), identifying 15 homologous genes. Additionally, we collected relevant literature and found limited reports of metalloproteases in plants. Specifically, only one M43-class protein (which induces cell death in tobacco leaves) and three M36-class proteins (involved in the degradation of plant chitinase) were reported.
Subsequently, we constructed phylogenetic analysis for these proteins. The results revealed that M43-class metalloproteases are widely distributed among fungi. However, in Fusarium, this class of proteins forms an individual cluster, while other fungi show more dispersed distribution. This suggests that the function of M43-class metalloproteases may have undergone further divergence specifically within Fusarium. Furthermore, the fact that M43 and M36-class metalloproteases are distributed in separate clusters implies significant functional differences between these two group proteins. Given this, our current understanding of Fg28's function remains incomplete. Additional studies involving other Fusarium species may provide more insights into its functional roles. We have incorporated this section into the Discussion and added a supplementary figure S2 and table S4. The corresponding methods have also been included in the Materials and Methods section for further clarification. All modifications are marked in red for clarity.
Round 2
Reviewer 1 Report
Most of my concerns were addressed. The only pending concern is related to the western blotting assays. Positive and negative controls are not optional for these assays, this is mandatory, It is the way these experiments must be conducted. I could see the unedited western blotting images to assess, please include them as supplementary material.
Most of my concerns were addressed. The only pending concern is related to the western blotting assays. Positive and negative controls are not optional for these assays, this is mandatory, It is the way these experiments must be conducted. I could see the unedited western blotting images to assess, please include them as supplementary material.
Author Response
Dear Editor and Reviewer 1,
We greatly appreciate your comments and suggestions that have helped us to significantly improve the manuscript for this revision. We have now made major revisions of our manuscript according to the reviewer 1 suggestions. We made response as below with our responses highlighted in Blue. The revised manuscript text shows major changes highlighted in red.
Reviewer 1
Most of my concerns were addressed. The only pending concern is related to the western blotting assays. Positive and negative controls are not optional for these assays, this is mandatory, It is the way these experiments must be conducted. I could see the unedited western blotting images to assess, please include them as supplementary material.
Response: Thanks for your suggestion. We have included the positive and negative controls of western blotting assays in Figure S2; the unedited western blotting images are provided in Figure S3. The relevant sections of the manuscript have been updated accordingly, including revisions in line 264, line 318, line 357 and lines 497-503.
Round 3
Reviewer 1 Report
The authors addressed my concerns.
The authors addressed my concerns.